# Star Polymers vs. Dendrimers: Studies of the Synthesis Based on Computer Simulations

**DOI:** 10.3390/polym14132522

**Published:** 2022-06-21

**Authors:** Piotr Polanowski, Krzysztof Hałagan, Andrzej Sikorski

**Affiliations:** 1Department of Molecular Physics, Lodz University of Technology, 90-924 Lodz, Poland; ppolanow@p.lodz.pl (P.P.); krzysztof.halagan@p.lodz.pl (K.H.); 2Faculty of Chemistry, University of Warsaw, Pasteura 1, 02-093 Warsaw, Poland

**Keywords:** dendrimers, dynamic lattice liquid, lattice models, polymerization, star-branched polymers

## Abstract

A generic model was developed for studies of the polymerization process of regular branched macromolecules. Monte Carlo simulations were performed employing the Dynamic Lattice Liquid algorithm to study this process. A core-first methodology was used in a living polymerization of stars with up to 32 arms, and dendrimers consisted of 4-functional segments. The kinetics of the synthesis process for stars with different numbers of branches and dendrimers was compared. The size and structure of star-branched polymers and dendrimers during the synthesis were studied. The influence of the functionality of well-defined cores on the structure and on the dispersity of the system was also examined. The differences in the kinetics in the formation of both architectures, as well as changes to their structures, were described and discussed.

## 1. Introduction

Star-branched macromolecules are of interest mainly because their size is smaller than that of linear polymers of the same molecular masses, and their viscosity behaves differently from linear chains, as it depends on the arm mass exponentially [1,2,3,4,5,6]. There are two main groups of synthetic methods for star synthesis: core-first and arms-first. In the first method, a multifunctional core is prepared in the first stage. Synthesis can be carried out using a large amount of cross-linker, and a large amount of the high conversion monomer is added to start growth arms [7,8,9,10,11,12]. If the reactivity of the cross-linker is higher than that of the monomer, the cores are synthesized at the beginning, and the rest of the monomer forms the arms of the star [13]. In the arms-first method, the linear polymers are synthesized first, and then a cross-linker is added and star cores are formed. The effectiveness of the cross-linking process depends mainly on the concentration of the cross-linker and solvent molecules [7,11,14,15,16,17]. Experiments [7,14,18,19,20] and simulations [21,22] have shown that a considerable fraction of the cross-linker also forms intrachain bonds which do not contribute to star core formation.

The theoretical interest in star-branched polymers is also significant [23]. Computer simulations of the properties of star-branched macromolecules were carried out using various methods [2,24,25,26,27,28,29]. Molecular dynamics simulations of isolated stars with various numbers of arms (between 6 and 50) that were rather short (the length of an arm was between 10 and 50) [30,31] were carried out, while the Cooperative Motion Algorithm [32] was used for simulation of star melts with up to 32 arms. Radke simulated isolated stars with 3, 8 and 16 arms, looking for their static properties, including hydrodynamic radius, which was crucial for observations of these macromolecules using size exclusion chromatography [33,34].

A star-branched polymer can be treated as a model of the simplest branched chain, but dendrimers can also be treated as very regular macromolecules. Dendrimers are polymers that are completely and regularly branched [35]. They consist of repeating units emanating from a central branching point [36,37]. They are typically characterized by functionality, which is usually *F* = 3 or 5, the spacer *m* (the number of segments between branching points) and the number of generations *g* (the number of shells containing junctions).

The synthesis of dendrimers is performed according to two main schemes: divergent and convergent, with increasing and constant reactions per growth step, respectively [38,39,40]. Dendrimers are desired because of their potential applications in nanotechnology, drug delivery, supramolecular science, catalysis, as coatings and biosensors, etc. [35,37,38,41,42,43]. The properties of dendrimers differ significantly from those of their linear and star-branched counterparts [44]. Dendrimers with a small number of generations form an entangled melt, while those with larger *g* form densely packed structures [45]. The number of chain entanglements is smaller than in linear chain systems; thus, their influence on viscoelastic properties is rather small, and the viscosity of dendrimers was found to be a nonmonotonic function of molecular weight [46,47,48].

Theoretical treatment of dendrimers is rather easy due to their regular structure and was mainly carried out using self-consistent mean field and renormalization group calculations [49,50]. The other theoretical approaches are computer simulations [51,52] employing Monte Carlo [53,54,55,56,57,58,59,60] and Molecular Dynamics [61,62,63,64,65,66] methods. The structure and viscoelastic properties of dendrimers studied by computer simulations showed the impact of the spacer length on the polymer properties [67,68]. The structure of dendrimers requires further studies, as it has a great impact on their properties in practical applications such as containers for drugs and nanomaterials [35,36,69,70,71,72].

In this paper, a comparison of the polymerization process of star-branched polymers and dendrimers was simulated by means of Monte Carlo simulations. In contrast to previous studies where initiator, cross-linker and monomer molecules were present [54], the prefabricated, well-defined cores were used in the synthesis. The focus was on the course of the synthesis process and on the scaling laws concerning the dimensions of the chain. The polymerization process was studied on a generic lattice model using the Dynamic Lattice Liquid model. The polymerization process was performed in bulk.

## 2. Simulation Method

In the presented model of the macromolecular systems, coarse-grained representation was used. The second simplification was the limitation of degrees of freedom by the introduction of a lattice approximation, and a face-centered cubic lattice (fcc) was employed for this purpose. An element of the system like a monomer, a mer in a growing chain (a polymer bead) was allowed to occupy one lattice point. Thus, the size of all of the above objects treated as beads was assumed to be identical. The status of objects could change as a result of the polymerization: each reaction step between beads that were neighbors (located in a distance of lattice constant) was irreversible and resulted in the formation of an unbreakable bond. The attempt of the reaction and of the modification of chains’ conformations takes place simultaneously with the probability of the attachment of a monomer 0.02. The Dynamic Lattice Liquid (DLL) model was used in simulations in order to provide the dynamics to the modeled system [73] with time expressed in algorithm steps (Monte Carlo steps). This model was described in detail in previous publications [20,34,35,71,72]; therefore, only a short description highlighting its main features is given here. In one Monte Carlo step, a field of randomly chosen unit vectors representing motion attempts was generated and assigned to all objects in the system. Only those attempts were successful which coincided in such a way that the sum of displacements along a closed path (loop) that included more than two molecules was equal to zero (condition of continuity). Each bead was then moved to a neighboring lattice site along this loop. All beads that did not contribute to these correlated loops stayed in their previous positions. This model of molecular systems is provided with a dynamics consisting of local vibrations and occasional diffusion steps. Within a longer time interval, this kind of dynamics leads to displacements of individual beads along random walk trajectories with steps distributed randomly in time. The most important advantage of this method is that it naturally takes into account changes in the local mobility of the system components and steric hindrances resulting from polymerization.

The Monte Carlo box was 100 × 100 × 100, and thus the system consisted of 10^6^ elements. The system was athermal (the excluded volume was the only interaction potential), and periodic boundary conditions were applied in all directions. At the beginning of a single simulation run, functional cores were randomly distributed in the system, while all the remaining lattice sites were filled with monomers while initiator molecules were omitted for simplicity. Simulated star-branched polymers were constructed using *f* = 4, 5, 6, 7, 8, 16 and 32 arms, and each system under consideration consisted of cores with the same functionality only. For stars with *f* = 4, a core consisted of one 4-functional monomer, but for a larger number of arms, next generations of 3-functional monomers must be added: for *f* from 5 to 8, the second generation of 3-functional monomers, for *f* = 16, the third generation, and for *f* = 32, the fourth generation. The concentrations of cores, i.e., the number of growing macromolecules, were selected in such a way as to obtain stars with approximately the same length of arms: the number of stars with *f* = 32 arms was twice that of stars with *f* = 16, four times that of stars with *f* = 8, etc. This made the concentration of cores 0.032% for stars with 4 arms, 0.016% for stars with 8 arms, etc. The concentration of multifunctional cores in the case of dendrimers’ synthesis was varied in the same way as for stars described above, i.e., 0.032, 0.01, etc. In all simulations of dendrimers, the spacer was set to 1, and the functionality of each monomer was 4. The length of arms in stars and the number of generations in dendrimers were not imposed on the model, and each simulation run lasted until exhaustion of the monomer molecules in the system (up to 10^6^ Monte Carlo steps).

## 3. Results and Discussion

### 3.1. Polymerization Kinetics

The kinetics of the synthesis processes for systems containing stars and dendrimers was studied in terms of the conversion of monomers, i.e., the factor related to the growing of macromolecules along with elapsed time. Figure 1a shows the conversion of monomers as a function of time for star-branched polymers with various numbers of arms in the log–log scale. In this case, one can distinguish two clear regimes for all cases, i.e., for the numbers of arms from 4 to 32. Figure 1b presents the derivative of monomer conversion as a function of time, i.e., the reaction rate. In the first regime, the increase in monomer consumption is linear, which means that the total reaction rate is constant, as can be seen in Figure 1b. In the second regime, the conversion of the monomer rapidly slows down, which corresponds to a rapid linear decline in the reaction rate, as presented in Figure 1b. These dramatic changes take place for the monomer conversion around 0.75–0.80, which is marked by the vertical dashed line in Figure 1a. This point indicates the time moment for which free-growing star macromolecules (no limitation of surrounding monomers) begin to compete with each other for available monomer molecules. This implies that the system, which is relatively diluted at first, becomes dense in terms of polymer concentration, and the location of this point very weakly depends on the number of branches. Differences in monomer conversion and the reaction rate between all macromolecular systems considered are small, which can be explained by the fact that the number of growing chains does not change in all star systems under consideration.

Figure 2 shows the monomer conversion and the reaction rate as a function of time for dendrimers. The star-branched polymers and dendrimers cannot be compared directly by means of the number of branches, but they can be compared when systems contain the same number of macromolecules (the same number of multifunctional cores). Figure 2a shows that the conversion of monomers is much faster than in the case of star-branched polymers, and after the time near 10^3^ Monte Carlo simulation steps, all monomer molecules reacted in all presented cases. The reaction rate presented in Figure 2b is slightly higher than for stars and increases in the initial period of time. This can be explained by the fact that for dendrimers, the number of active ends increases sharply as the dendrimer grows, while it remains constant in the case of stars. The overall monomer reaction rate for dendrimers slowly increases in the initial regime and then accelerates until the monomer is completely depleted.

Obviously, these differences in the monomer conversion for stars and dendrimers lead to differences in the growth dynamics and influence their scaling properties. Dynamic scaling of cluster size is usually discussed using the Smoluchowski coagulation equation [74]. It describes the cluster size distribution *n_l_* for dilute systems, where only binary collisions are relevant. For long time scales and large clusters, the number of clusters formed by *l* elements exhibits dynamic scaling, and the cluster size distribution approaches the limit form [75]:*n_l_* ≈ *N*^−2^ Φ(*N*/*l*),(1)
where *N* = *N*(*t*) is related to the number-average cluster size, and Φ(*N*/*l*) is a universally time-independent cluster size distribution, which characterizes the aggregation mechanism. Combining the solution of Equation (1) with the fractal scaling relationship leads to:*N*(*R*) = *k*_0_*R^d^_f_*,(2)
where *k*_0_ is a prefactor, *R* is a characteristic size, *d_f_* stands for fractal dimension and *N*(*R*) is the number of particles aggregated in the cluster. The mean cluster size defined in Equation (2) can be also obtained as a function of time [76]:*N*(*t*) ≈ *t^z^* ≡ *t*^1/(1−*λ*)^,(3)
where *λ* is the van Dongen and Ernst homogeneity exponent [77] that describes different growth kinetics induced by the object–object interaction. It should take the value 0 for the diffusion-limited cluster aggregation and 1 for the reaction-limited cluster aggregation. In the presented case, the role of *N*(*t*) plays the number-averaged mass *M_w_* of star or dendrimer (number of mer units).

Figure 3a presents the changes of the average mass *M_w_* of the star-branched chain as a function of time *t*. Three regimes can be distinguished for all cases. The first is the initial one, where the growth of the star is slow. In the second regime, the mass increases significantly. In the third, the growth becomes slow again due to the high polymer density and the competition between stars in acquiring the remaining monomer. The influence of the arms number becomes visible if one considers the average mass of one arm *M_w_*/*f,* which is presented in Figure 3b. Three regimes of behavior can also be distinguished in this case. In the first one, the increase of an average arm mass is slow, and a higher number of arms increase their mass. In the second regime, the arm mass increase does not depend on the number of arms and increases with time approximately as *t*^0.84^. This exponent is close to the reaction-limited cluster aggregation scenario characterized by exponent 1. In the third regime, the mass of arms stabilizes and does not depend on the number of arms—it scales with time as *t*^0.045^. In this regime, the mean size of an arm does not scale according to Equation (3) because the growth of the macromolecule takes place at very high concentrations, while Equation (3) describes the system at low and moderate concentrations.

The changes of dendrimer polymer mass over time are presented in Figure 4. The initial period is common for all cases with a slow increase of mass. Then, a rapid mass increase is observed, which lasts until the monomer is finally depleted. This increase is considerably stronger than in the case of star-branched polymers with the exponent of 2.30.

The kinetics of polymer growth during the polymerization process can be characterized by basic polymerization indicators such as *M_w_*, weight average mass *P_w_* and dispersity *P_w_*/*M_w_*. In this representation, the monomer conversion degree usually plays the role of non-linear time.

Figure 5 shows the number-averaged mass *M_w_*, the weight-averaged mass *P_w_* and the dispersity *P_w_*/*M_w_* for the star-branched systems under consideration in a log–log plot. After a short initial period, where the consumption of the monomer is low and stable, the increase is almost linearly proportional to the monomer conversion in a log–log scale. It was shown that in real experiments (ATRP polymerization), this dependency is the same [78]. The dispersity increases in the initial period, and the more arms the stars have, the lower the dispersity is. In general, the star-branched macromolecules with well-defined cores exhibit very small dispersion (below 1.07).

Figure 6 shows the number-averaged mass, the weight-averaged mass and the dispersity for the dendrimer systems under consideration in a log–log plot. In this case, the changes of *M_w_* and *P_w_* are similar to those for star-branched polymers, as discussed above. The dispersity almost does not depend on the number of macromolecules and increases with monomer conversion. This parameter is slightly higher than for star-branched polymers, and it remains rather low.

In order to obtain insight into details of dispersity values in both discussed cases, total mass distributions must be examined. Figure 7 presents the polymer mass distributions of macromolecules at the end of the polymerization process for all systems under consideration fitted with a Gaussian function. One can see that, in spite of small differences in the dispersity of stars and dendrimers (Figure 5 and Figure 6), the distributions of mass for dendrimers are considerably broader. For stars, the width of the distribution is almost constant, while for dendrimers it decreases when the number of macromolecules increases.

### 3.2. Molecular Topology during the Polymerization Process

The basic quantities used to characterize the geometric properties of a macromolecule are the radius of gyration and the hydrodynamic radius. In the presented cases, for low dispersion, one can define the mean-squared radius of gyration < *R_g_*^2^ > and the mean hydrodynamic radius < *R_h_* > as follows:< *R_g_*^2^ > = 1/*M* ∑_j_ *R*^2^*_g,j_*,(4a)
*R*^2^*_g,j_ =* 1/*N_j_* ∑_i_ (**r***_i,j_*–**r***_cm,j_*)^2^,(4b)
where *M* is the total number of macromolecular objects in the system (stars or dendrimers), *N_j_* is the number of beads in a given macromolecule and **r***_cm,j_* is the location of its center of mass.
< *R_h_* > = 1/*M* ∑_j_ *R_h,j_*,(5a)
1/*R_h,j_ =* 1/2 ∑_m_ ∑_n_ 1/**r***_mn,j_*,(5b)
where **r***_m,n_* is the distance between two different beads *m* and *n* in the molecule *j*.

The radius of gyration and hydrodynamic radius were calculated for both architectures studied according to Equations (4) and (5). The < *R_g_* >/< *R_h_* > ratios for both stars and dendrimers are presented in Figure 8. The results clearly reflect this ratio for stars, i.e., the ratio decreases with the number of arms, although it differs from the experimental data [79]. For small dendrimers, the ratio does not depend on the number of macromolecules in the system, while for larger ones it diverges.

The shape of macromolecules is usually described by means of the asphericity factor, which is defined as
(6)$$A_{}=\frac{{\left(\lambda_{2}-\lambda_{1}\right)}^{2}+{\left(\lambda_{3}-\lambda_{1}\right)}^{2}+{\left(\lambda_{2}-\lambda_{3}\right)}^{2}}{{\left(\lambda_{1}+\lambda_{2}+\lambda_{3}\right)}^{2}}$$
where *λ_i_* are the components of the squared radius of gyration along the principal axes of inertia [80]. Figure 9a presents the mean asphericity factor *A* determined according to Equation (6). The shape of stars becomes more spherical when the number of arms increases, which is consistent with other findings [2]. In the case of dendrimers, there is a minimum on the curve, which was already found by experiments [40]. Polymer density profiles for both architectures are presented in Figure 9b. Polymer density decreases very slowly, which suggests the presence of a dense core and a rapid decrease for longer distances. In their theoretical considerations, Daoud and Cotton [81] predicted two regimes inside stars: an unswollen one with slope −1 and a swollen one with slope −4/3. One can conclude that it is impossible to localize the regions with such slopes on our curves.

Another important aspect of studying the topology of macromolecular objects is that their geometry changes with the synthesis progress with respect to both time and total mass (total number of beads *N*). Typically, for stars, the mass scaling is expressed in the following form:< *R*^2^ > ≈ *N*^2*ν*^ ≈ *N*^2/*d*^*_f_*,(7)
where *R* is a characteristic dimension of the entire object (e.g., radius of gyration), *ν* the scaling exponent and *d_f_* is the object fractal dimension.

Figure 10 shows the changes in the mean-squared radius of gyration as a function of star mass and time. A scaling behavior of < *R_g_*^2^ > was found not to be related to the number of arms. The exponent 2*ν* decreases from 1.01 to 0.83, as the number of arms increases, which means that the fractal dimension becomes higher (from 1.98 to 2.41) with the number of arms. The exponent 2ν from Equation (7) for dense polymer melts is expected to be close to 1 [82], while lower values are predicted for dense collapsed structures. The size growth exponent is similar for all investigated numbers of arms.

In the case of dendrimers, the scaling of chain size is more complex. Using Flory theory, it was shown that under good solvent conditions, the radius of gyration should scale according to the following formula [50,53,54]:*R*_g_ ≈ *N*^1/5^ [(*g* + 1)*m*]^2/3^,(8)
where *g* is the number of generations and *m* is the length of the spacer. Using simple arguments based on the properties of a geometric series (in investigated cases, the beads were 4-functional, with *m* = 1), it can be stated that the number of beads *N* corresponding to the generation *g* is given by the following formula [49,53]:*N* = 1 + 2(3*^g^*^+1^ − 1),(9)

By solving Equation (9) with respect to the number of generations *g,* one can easily find *g* corresponding to number (mass) *N* of the dendrimer′s elements.

*g* = log_3_((*N* + 1)/2) − 1,(10)

Figure 11 shows the changes in the number of generations *g* calculated according to Equation (10) as a function of time of synthesis and the total mass of a macromolecules. In the initial period, the estimated number of generations does not depend on the number of growing macromolecules. Moreover, it is clear that the increased scales with time and the scaling exponent are found to be close to 0.47. Then, well below 10^3^ Monte Carlo steps, *g* clearly depends on the number of cores (macromolecules) in the system. At the lowest initiator concentration (the smallest number of cores), almost 8 generation dendrimers were achieved, while for the highest number of cores, the number of generations does not exceed 6 (one must remember that the polymers studied are not the same length and the mean number of generations is studied here). The dependence of the number of generations on the total dendrimer mass is presented in Figure 11, showing no differences in the investigated cases.

To verify the validity of these theoretical considerations, a reduced radius of gyration *R_g_*/((*g* + 1) − 3)^2/5^ was calculated based on Equation (8). Figure 12 presents it as a function of the total number of beads *N*. The scaling exponent was found to be 0.21 for all cases under consideration, and very close to the value 1/5 predicted by Flory theory given by Equation (8). This is consistent with the simulation results obtained for dendrimer lattice models at good solvent conditions [53,83]. The deviation from this value is observed when macromolecules become larger and interchain interactions become an important factor (as the polymer concentration approaches 1), which makes the dependence stronger. Figure 12 indicates that the size of dendrimers *R_g_*^2^ does not scale ideally with the number of beads *N* but also with the topological parameter *g*. This means that dendrimers obtained in the simulated synthesis process do not possess a well-defined fractal dimension, and thus, are not true fractal objects (for a model dendrimer, the fractal dimension should be close to 3). This behavior is consistent with Flory’s theory and was observed in other simulation studies [53].

The visualization of a single macromolecule clearly shows the differences between the studied polymer architectures. Figure 13 presents examples of single star conformations with a small number of arms (*f* = 4) and a large number of arms (*f* = 32) and the corresponding dendrimer structures. One can observe the formation of a dense core for the star with a high number of arms. The difference between stars and dendrimers is also striking as the entire dendrimer must be considered as a densely packed globule.

## 4. Conclusions

The simulations of a generic model of star and dendrimer polymers were performed by means of the Monte Carlo method. The Dynamic Lattice Liquid model was employed for studies of the bulk polymerization process and characterization of structural properties. The model system contained well-defined cores where the growing of chains (the attaching of multifunctional monomers) takes place. The polymerization was carried out in bulk to the exhaustion of monomers with no solvent present.

The simulation results clearly indicated that the process of synthesis of highly branched macromolecules like stars and dendrimers using well-defined cores led to macromolecular structures with well-defined properties. It was shown that, for both macromolecular architectures and for all discussed cases (number of arms for stars and number of macromolecules for dendrimers), the dispersity was found to remain very low during the entire process of polymerization. Thus, this parameter was found to be considerably lower than for systems where cores were formed from initiator and cross-linker molecules simultaneously with the process of polymerization, i.e., attaching monomers to growing macromolecules. The mass of stars and dendrimers was rather comparable. The kinetics of the synthesis process for stars and dendrimers was compared, showing considerably faster polymerization of dendrimers, evidently because of a higher (and growing) number of points where a macromolecule could attach monomers. The reaction-limited cluster aggregation scenario was found to more suitable for stars. The size and structure of star-branched polymers and dendrimers during synthesis were found to be different, and the scaling of polymer size with its total number of beads was different too.

## Figures and Tables

**Figure 1 polymers-14-02522-f001:**
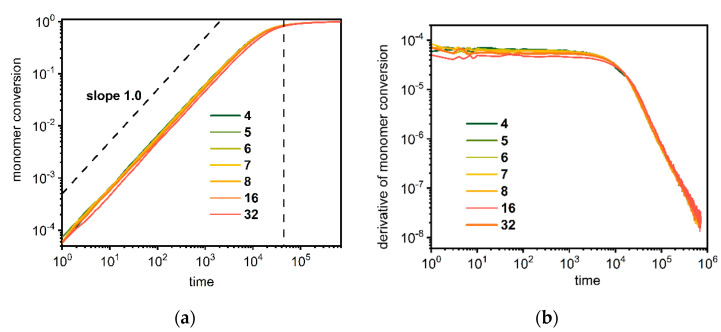
Conversion of a monomer (**a**) and its derivative (**b**) as a function of time during polymerization of star-branched polymers. The number of arms is given in the (**b**) legend.

**Figure 2 polymers-14-02522-f002:**
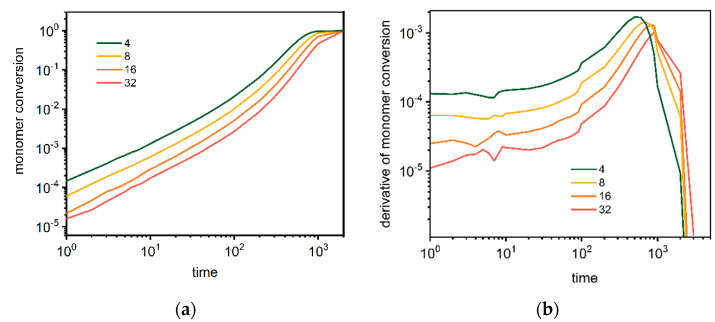
Conversion of a monomer (**a**) and its derivative (**b**) as a function of time during polymerization of dendrimers. The color indicates the number of macromolecules in the system (see text for details).

**Figure 3 polymers-14-02522-f003:**
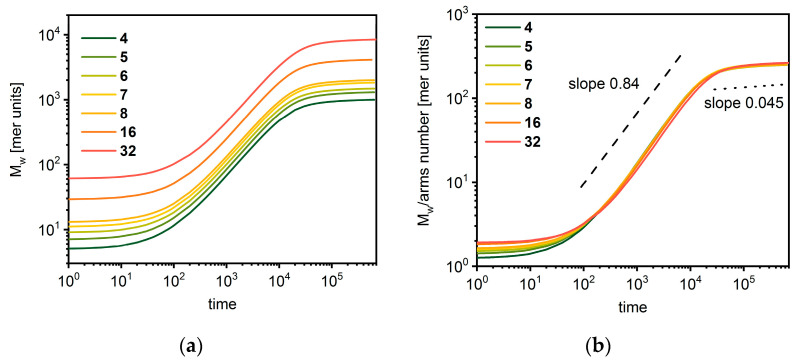
Average total mass (**a**) and average mass of arm (**b**) of star-branched polymer as a function of time. The number of arms is given in the legend.

**Figure 4 polymers-14-02522-f004:**
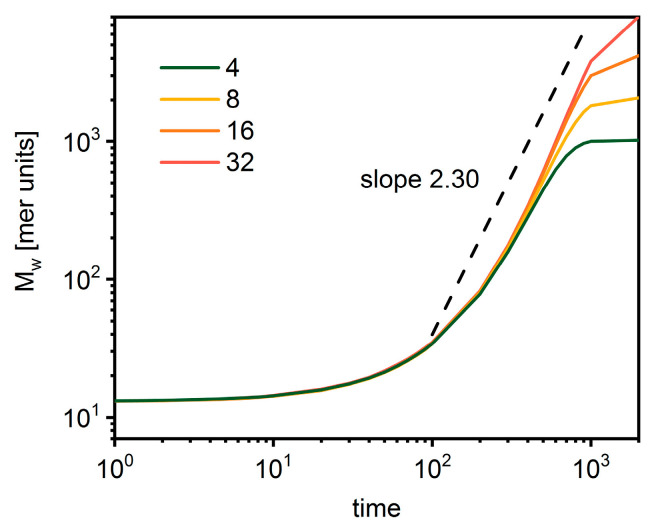
The average total mass of the dendrimer as a function of time. The color indicates the number of macromolecules in the system (see text for details).

**Figure 5 polymers-14-02522-f005:**
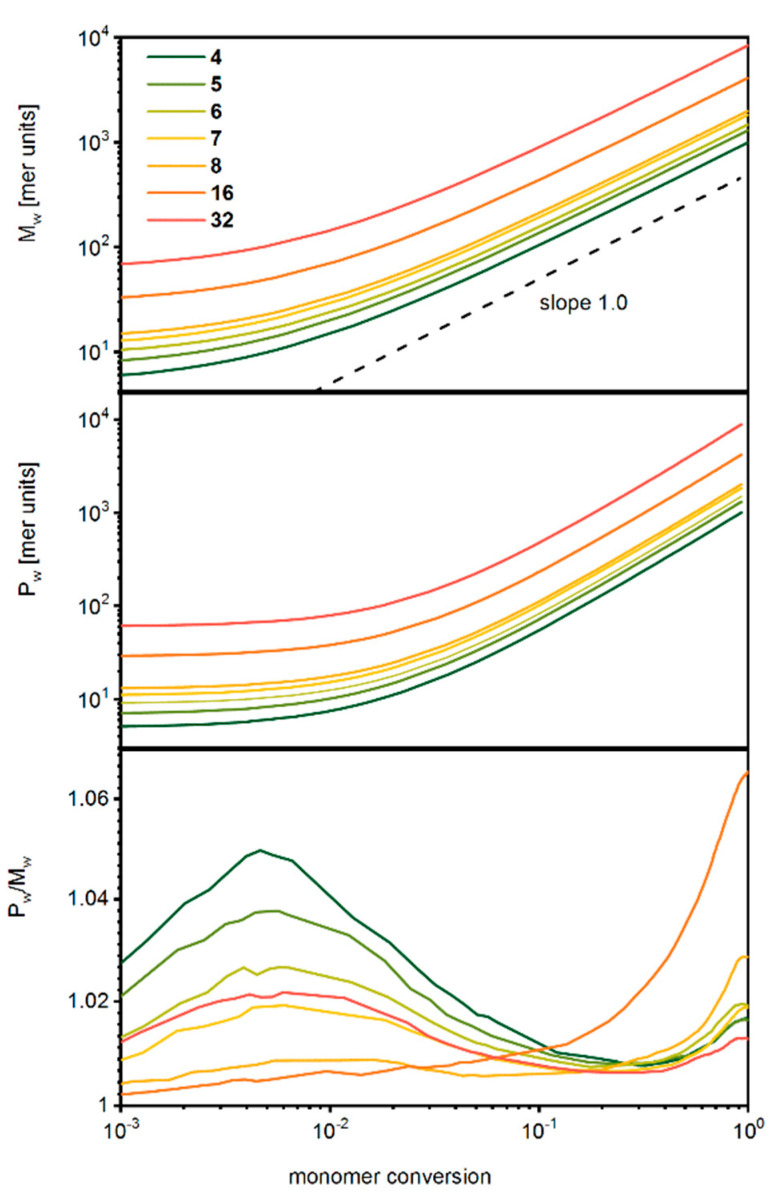
Number-averaged mass (upper panel), weight-averaged mass (middle panel) and dispersity (lower panel) of star-branched polymers as a function of the monomer conversion. The number of arms is given in the legend.

**Figure 6 polymers-14-02522-f006:**
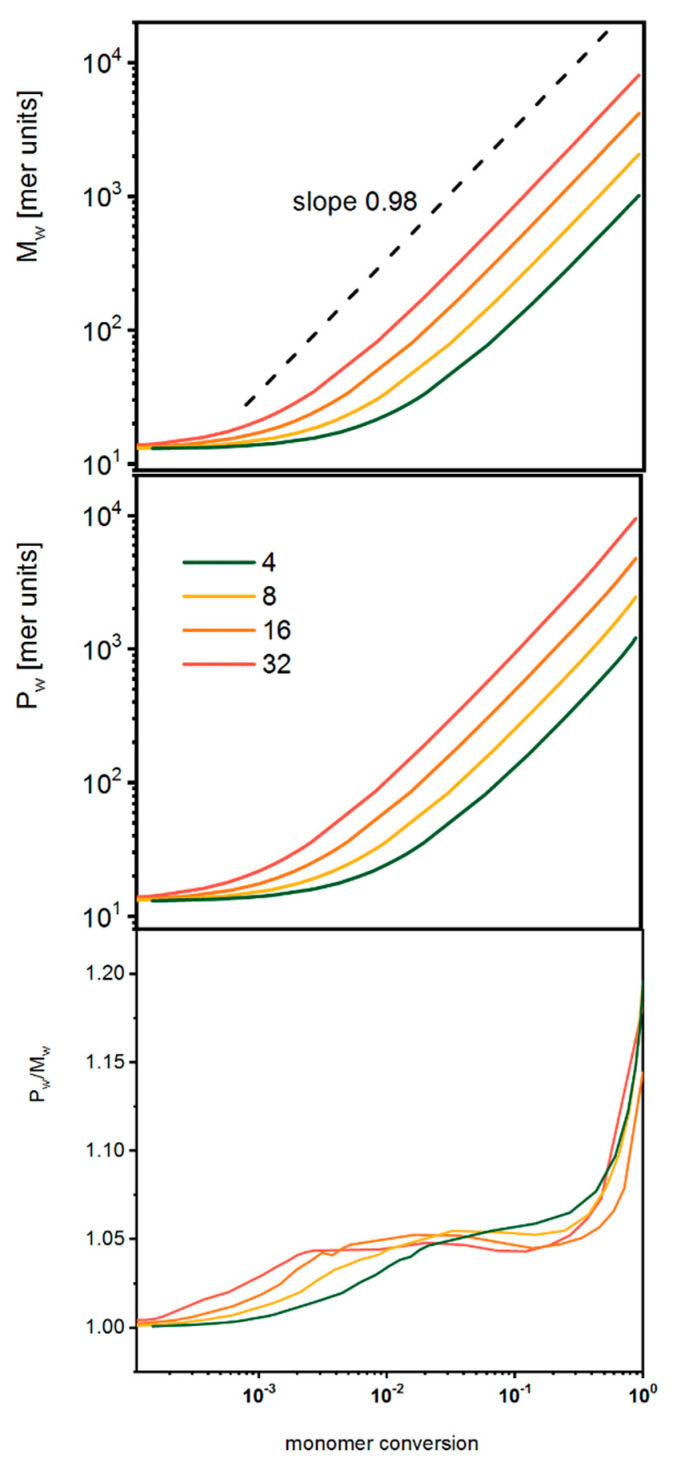
Number-averaged mass (upper panel), weight-averaged mass (middle panel) and dispersity (lower panel) for dendrimers as a function of the monomer conversion. The color indicates the number of macromolecules in the system (see text for details).

**Figure 7 polymers-14-02522-f007:**
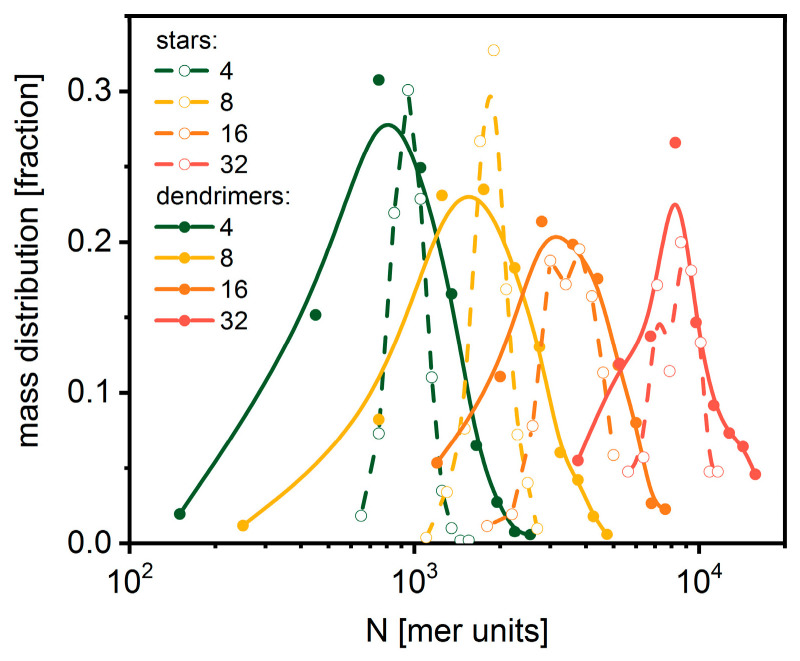
The mass distribution for star-branched polymers (open symbols) and dendrimers (solid symbols). The lines indicate Gaussian fits.

**Figure 8 polymers-14-02522-f008:**
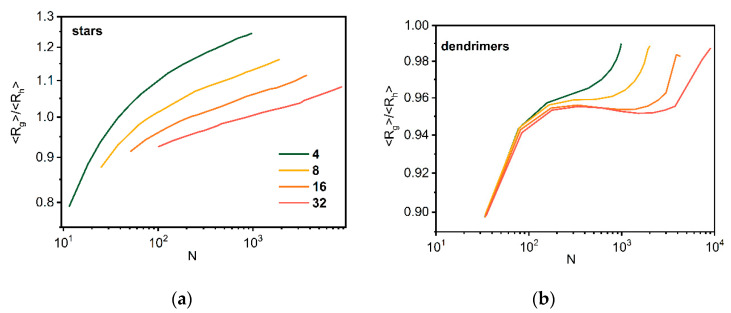
The ratio < *R_g_* >/< *R_h_* > as a function of total number of beads *N* for stars (**a**) and dendrimers (**b**). The color indicates the number of macromolecules in the system (see text for details).

**Figure 9 polymers-14-02522-f009:**
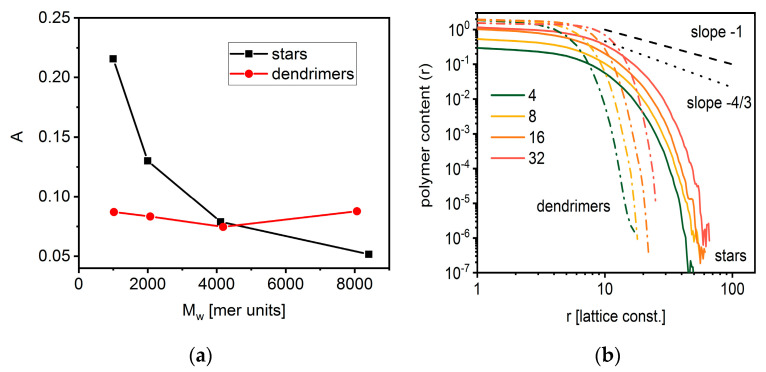
Asphericity as a function of polymer mass (**a**) and density profiles (**b**) for star-branched chains and dendrimers (see text for details).

**Figure 10 polymers-14-02522-f010:**
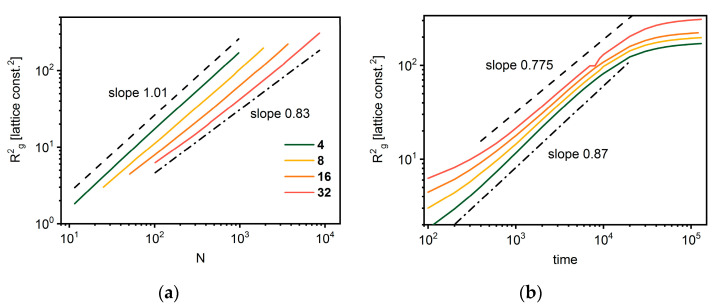
The mean-squared radius of gyration < *R_g_*^2^ > of star-branched chains as a function of total number of beads *N* (**a**) and as a function of time (**b**). The colors indicate the number of arms.

**Figure 11 polymers-14-02522-f011:**
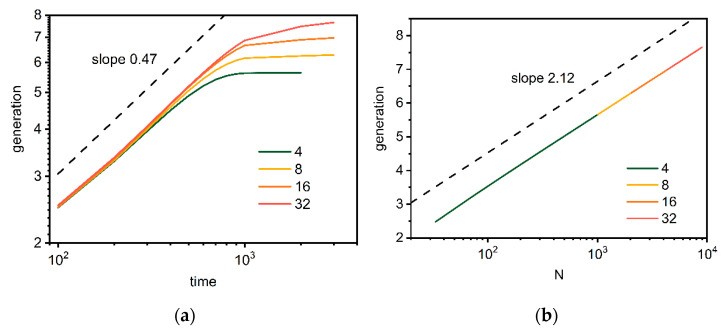
The number of dendrimer generations g as a function of time (**a**) and total number of beads *N* (**b**). The color indicates the number of macromolecules in the system (see text for details).

**Figure 12 polymers-14-02522-f012:**
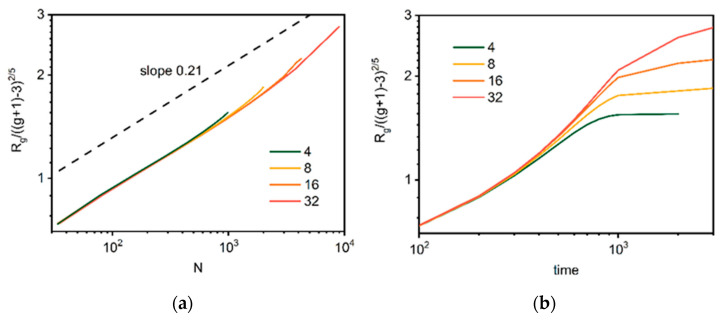
The reduced radius of gyration for dendrimers *R_g_*/((*g* + 1) − 3)^2/5^ as a function of total number of beads *N* (**a**) and time (**b**). The color indicates the number of macromolecules in the system (see text for details).

**Figure 13 polymers-14-02522-f013:**
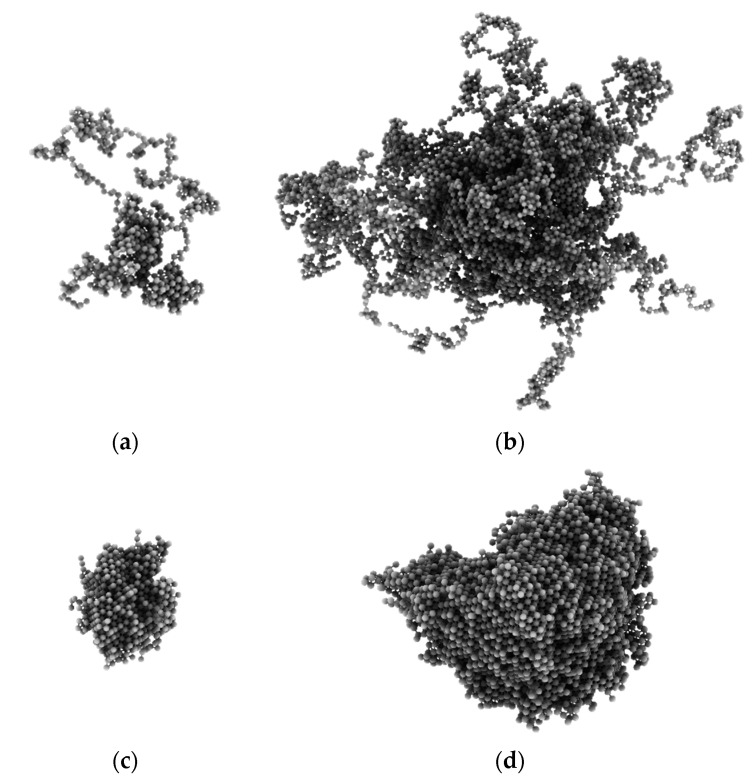
Snapshots of randomly selected star-branched polymers and dendrimers. The stars consisted of *f* = 4 (**a**) and *f* = 32 (**b**). The dendrimers (**c**,**d**) represent systems containing the same number of cores (macromolecules) as those with stars presented in the upper row.

## Data Availability

The data that support the findings of this study are available from the corresponding author upon reasonable request.

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
