# Peer review of "Star Polymers vs. Dendrimers: Studies of the Synthesis Based on Computer Simulations"

_polymers, 2022, doi:10.3390/polym14132522_

Round 1

Reviewer 1 Report

The manuscript aims to employ Monte Carlo methods and DLL algorithm to model the formation of star and dendrimer polymers during core-first synthesis. In case of star polymers, systems with up to 32 arms, and in case of dendrimers the 4-functional segments were used. The authors presented the total monomer conversion rate, i.e. Mw, during the synthesis and the structure of final aggregates was described by radii-of-gyration and related hydrodynamic radius.

Remarks

The term coarse-grained model is misleading in context of this paper and authors description. Instead I would use coarse-grained representation or generic model. The difference is that CG model is something more and usually requires mapping of properties from different scales including building some extra potentials etc. and targeting to real systems.

I would strongly recommend to updating the literature in the second paragraph of the Introduction, where the references are very old. The newest is from 2001. Better situation is for dendrimers in lines 61-68.

Simulation Method section

I am missing any info about setting the simulation and determining the proper simulation time and size of the simulation box. The authors mention in line 98 the different regimes of mean-squared displacement. Is that the state of their model?

Also, I would add better description of dendrimers model. For example, how many generations the dendrimer model achieved during the modeling. This is not clear to me.

This info can be put to SI if necessary that is completely missing.

Results

Figure 1 -and corresponding comments in lines 133-137. Although the figure is in log scale, it is still quite surprising to me that the conversion rate does not depend on the # of arms in start polymer. If applicable. Could you comment more on that for example in terms of Daoud-Cotton model?

Figure 3b – the slope in the mid part of the curve should be very close to 1, 0.84 in your case. Could you comment more the deviation from theoretical prediction? For example, I am missing the info, if you applied any correction to Smoluchowski coagulation equation that can possibly influence the slope.

Figure 4 – I am missing figure, to be able to compare with its counterpart in Figure 3b.

Figure 5 – please add the slope to the figure, ate least for Mw.

Figure 7 – please add the legend or description into the figure.

Figure 8 – change dendrymers to dendrimers. Furthermore, according to the literature, some polymers decrease the ratio of Rg and Rh while increasing the number of generations. Your case is the opposite one. Please add the comment to which systems your behavior fits.

Line 250-253: The authors attempt to describe the shape and geometry of the start and dendrimer only by ratio of Rg and Rh. I think that it is not enough and more suitable variables from the shape descriptors group exist for that. For example, shape anisotropy, asphericity, acylindricity etc. Moreover, the density distribution of chains from centre-of-mass of the core would be also helpful to obtain deeper insight into the system.

Finally, I am missing stronger link to real synthesis that would be necessary to evaluate the usefulness and precision of the model itself.

Questions

Equation 3: should not be the z in the superscript?

Equation 6: According to athermal conditions and Flory theory, the v=0.588. Your slope should be 1.176. Yours fits in between 0.83- 1.01. Can you please explain that?

Howe many generations your dendrimer achieved during the synthesis?

Can you link, somehow, your results with Daoud-Cotton model scaling?

Decision

According to above mentioned remarks and questions, I think that major revision is needed for this manuscript. Mainly, I would add the shape descriptors and density profiles and link the results with real synthesis. Also, the better connection with scaling models (Daoud-Cotton, Smoluchowski coagulation model) would be helpful. After processing these remarks and questions, the manuscript is suitable for publishing in Polymers journal.

Reviewer 2 Report

The article describes the computer simulation of the polymerization process of regular branched macromolecules - dendrimers. The authors tried to simulate ste-by-step synthesis of 4,8,16,32 - branches of dendrimers. The topic is new and interesting for wide reader. The results are correctly done and described in details using more than 70 references. The figures illustrate the data. The conclusions are supported by results. In general, the article is well written and can be published. I did not find any serious mistakes to mention. As for English, I am not native speaker, for me English is O.K.

Reviewer 3 Report

It was a good manuscript about the application of computer simulations for comparing different structural properties of Star polymers and dendrimers. Here are some comments on this study that should be considered before publication:

1-     There are some grammatical mistakes in the text that should be corrected.

2-     Please refer to the equations in the main text.

3-     “Combining solution of Eq. (1) with the fractal scaling relationship” it seems this sentence is incomplete.

4-     Please add the unit of axis in figures.

5-     Please explain more about the results of figure 5.

6-     “dendrimer” is written wrong in figure 8-b.

7-     “This behavior is in agreement with Flory’s theory and was observed in other simulation studies” please add more references for this sentence.

8-     The conclusion part could be improved.

9-     The last paragraph of introduction should also be improved.

Round 2

Reviewer 1 Report

The authors have answered all my questions and significantly changed the manuscript. Therefore, I have no other questions or comments and recommend to accept the paper in present form.

Author Response

The authors would like to thank the reviewer.

Reviewer 3 Report

Please refer to Eq.2 and Eq.6 in the main text.
